# Preparation of Paclitaxel-Encapsulated Bio-Functionalized Selenium Nanoparticles and Evaluation of Their Efficacy against Cervical Cancer

**DOI:** 10.3390/molecules27217290

**Published:** 2022-10-27

**Authors:** Soumya Menon, Santhoshkumar Jayakodi, Kanti Kusum Yadav, Prathap Somu, Mona Isaq, Venkat Kumar Shanmugam, Amballa Chaitanyakumar, Nagaraj Basavegowda

**Affiliations:** 1Department of Chemistry, Indian Institute of Technology, Roorkee 247667, India; 2Department of Biotechnology, Saveetha School of Engineering, Saveetha Institute of Medical and Technical Science (SIMATS), Chennai 602105, India; 3Department of Biotechnology, Karunya Institute of Technology and Sciences (Deemed to be University), Karunya Nagar, Coimbatore 641114, India; 4Department of Biotechnology & Bioinformatics, Kuvempu University, Jnana Sahyadri, Shankaraghatta, Shivamogga 577451, India; 5School of Bio-Sciences and Technology, Vellore Institute of Technology, Vellore 632014, India; 6Department of Biotechnology, University Institute of Engineering and Technology, Guru Nanak University, Hyderabad 500085, India; 7Department of Biotechnology, Yeungnam University, Gyeongsan 712-749, Korea

**Keywords:** *Mucuna pruriens* seed extract, chitosan, paclitaxel, selenium nanoparticles, flow cytometer, anticancer, cervical cancer

## Abstract

The potentiality of nanomedicine in the cancer treatment being widely recognized in the recent years. In the present investigation, the synergistic effects of chitosan-modified selenium nanoparticles loaded with paclitaxel (PTX-chit-SeNPs) were studied. These selenium nanoparticles were tested for drug release analysis at a pH of 7.4 and 5.5, and further characterized using FTIR, DLS, zeta potential, and TEM to confirm their morphology, and the encapsulation of the drug was carried out using UPLC analysis. Quantitative evaluation of anti-cancer properties was performed via MTT analysis, apoptosis, gene expression analysis, cell cycle arrest, and over-production of ROS. The unique combination of phytochemicals from the seed extract, chitosan, paclitaxel, and selenium nanoparticles can be effectively utilized to combat cancerous cells. The production of the nanosystem has been demonstrated to be cost-effective and have unique characteristics, and can be utilized for improving future diagnostic approaches.

## 1. Introduction

Nanomedicines are emerging as a significant class of therapeutics. The most encouraging aspect of using nanoparticles in therapeutics is their capability to target or accumulate at the site of targeted tumour tissues with negligible toxicity. The nanomaterial, owing to its size, permeates through the kidney or liver due to surface decorations, and the cells diffuse through the blood vessels and vasculature of tumour tissues, thereby extending its presence in the blood circulation and this is explained through the phenomenon of EPR (enhanced permeation and retention). Moreover, nanoparticles have high surface area to volume proportions [1], yielding high adsorbing properties [2]. On this basis, nanoparticles are adsorbed with therapeutic drugs [3], phytochemicals with anticancer properties [4], imaging agents [5], or target-specific genes or peptides as targeting ligands to the cancer receptor cells [6].

Selenium (Se) is a trace element with broad pharmacological capabilities and physiological functions. It is an essential component of numerous anti-oxidant enzymes like phospholipid hydroperoxide, glutathione peroxidase, and thioredoxin reductase. It is a micronutrient that plays various roles in human health, improving cardiovascular health or immunomodulatory functions, constraining cancer progression, preventing neurodegeneration, and regulating the thyroid hormone metabolism [7,8]. Selenium supplements traditionally used for cancer treatment have disadvantages of high toxicity or low absorption. Therefore, developing novel systems as transporters of selenium compounds that would promote bioavailability and permit its controlled release in the organism has become a significant requirement in this field. The characteristic features of Se can be explained through its antioxidant properties by protecting against oxidative stress generated by oxidative radicals or pro-oxidant properties by promoting a strong generation of ROS via the redox cycle and causing oxidative stress to cancer-infected cells [9,10,11,12]. The mechanism of chemotherapeutic action of SeNPs can be explained through the over-expression of antioxidant enzymes that induce the generation of ROS, eventually activating a sequence of events, such as induction of the apoptotic pathway and mitochondrial dysfunction that stimulates the release of cytochrome C, thereby inducing the activation of caspase cascade, cell cycle arrest, and DNA fragmentation [13]. The latest investigations report that the chemotherapeutic activity of SeNPs has been successful against various malignant cells, such as human cervical carcinoma (HeLa) [14], breast cancer (MCF-7) [15], human hepatocyte (HepG2) [16], human melanoma (A375), and lung cancer (A549) cells [17]. However, the major problem associated with the application of SeNPs as anticancer activity is reported to be undesired toxicity against healthy cells [10]. Thus, we need to overcome this undesired toxicity without compromising its anticancer activity. The functionalization of the NP surface with biopolymers is one of the best techniques reported, to date, for improving biocompatibility with reduced aggregation.

Thus, chitosan is used for the functionalization. Chitosan is a cationic hydrophilic linear polysaccharide composed of an acetylated unit (N-acetyl-d-glucosamine) and a deacetylated unit (β-(1–4)-linked D-glucosamine). These structural characteristics of chitosan confer membrane permeability, biocompatibility with tissues, and mucoadhesive properties. Additionally, they possess a pH-dependent profile, where the primary amines become positively charged by protonation in a low pH environment. The material becomes easily soluble in the medium, thereby assisting in the encapsulation of drugs. At high pH, chitosan becomes insoluble, as the primary amines are in deprotonated conditions. Therefore, pH plays a major role in this delivery system as the body may have various regions with significantly different levels of pH. In the stomach, chitosan can easily permeate the acidic surroundings without degradation by digestive enzymes and remain stable at physiological conditions, due to its mucoadhesive properties [18,19,20].

Moreover, SeNPs might also be used for drug delivery applications, thereby providing a synergistic therapeutic effect against cancer. Paclitaxel (PTX) is a compound derived from the bark of *Taxus brevifolia* [21] and is a well-established anticancer drug that can prohibit the mechanism of mitosis or polymerize the β-microtubules, thereby arresting the cell cycle and causing cell death [22]. Regardless of its exclusive mechanism and effectiveness, it lacks properties such as high therapeutic index and adequate solubility [23]. To improve its efficiency of solubility, PTX was conjugated with biosynthesized selenium nanoparticles modified with chitosan. In the present study, the Box–Behnken design was selected to optimise PTX-chit-SeNPs and was compared with the biosynthesized SeNPs for its anticancer progression activities and its cytotoxicity analysis against HeLa cells. Alternative formulations, such as nanoparticles, liposomes, self-emulsions, and micelles can aid in improvising the effective delivery of PTX to tumour cells with low toxicity to healthy cells. Among these, nanoparticles that are functionalized with phytochemicals from seed extract as novel agents can provide a promising delivery system with the advantage of the EPR effect for passive targeting of PTX. The results of this study suggest that this nano-combinational system may be a promising low toxicity alternative for cancer treatment; however, further analysis of the in vivo or ex vivo studies can still be reviewed in future studies.

## 2. Material and Methods

### 2.1. Materials

The seeds of *Mucuna pruriens* were bought from a local drug store. Sodium selenite, paclitaxel (PTX), chitosan with 90% deacetylation (MWCO 55 kDa), methanol, and acetonitrile (HPLC grade) were purchased from Hi-media. All of the chemicals were analytical grade and no further purification was required.

### 2.2. Cell Culture

HeLa cells were purchased from NCCS, Pune, India. The cells were maintained using DMEM, 10% (fetal bovine serum) and (1%) penicillin–streptomycin antibiotic and were incubated in a cell culture incubator with 5% CO_2_ at 37 °C in a humidified atmosphere. Chemicals such as PI (propidium iodide) reagent, RNase, Invitrogen Annexin V/FITC dead cell kit by Thermo Fisher Scientific (Waltham, MA, USA), 3-(4,5-dimethylthiazolyl-2)-2,5-diphenyltetrazolium bromide MTT reagent, DCFDA-cellular ROS Assay kit, PBS (10X at pH 7.4) were purchased from Hi-Media, and a Primescript RT reagent kit was purchased from Takara Bio (Kusatsu, Japan).

### 2.3. Preparation of Biosynthesized Selenium Nanoparticles

The biosynthesis of SeNPs using seed extract of *Mucuna pruriens* and its standardization was published in our previous study [24]. In brief, the seeds were double-washed with double-distilled water (Mili-Q), shade-dried for 5–6 days, and the dried seeds were crushed, weighed, and then added to 100 mL of Mili-Q. The liquid was kept in ta water bath at 80 °C for 45 min, and filtered using Whatman 42 filter paper. The filtrate was stored at 4 °C for further use. A ratio of 9:1 was maintained for precursor and extract concentration. The reaction continued till the colorless solution changed to the dark brown reaction mixture. The synthesized nanoparticles were further purified with double-distilled water and ethanol solution; this helps in the removal of excess phytochemicals or untreated sodium selenite from the SeNPs solution. The pellet was then dialyzed against distilled water for a period of 12 h. The purified nanoparticles were lyophilized for further characterization and storage.

### 2.4. Preparation of Chitosan-Modified and Paclitaxel-Loaded Selenium Nanoparticles

The chitosan solution of 1% concentration (*w*/*v*) was prepared in glacial acetic acid where the solution was kept in a magnetic stirrer for 2 h until it was completely dissolved, followed by the addition of 1% of the biosynthesized SeNPs with continuous stirring at RT for 10–12 h. After the12 h of stirring, paclitaxel in DMSO (at the required concentration) was added drop-wise, at 1 mL/min, to the chitosan-coated selenium nanoparticles and kept overnight for continuous stirring. The PTX-chit-SeNPs were then centrifuged at 10,000 rpm for 15 min, the pellet obtained was washed with Mili-Q water and the purified solution was then lyophilized. The moisture-free samples were then stored at room temperature for further analysis and characterization.

### 2.5. Entrapment Efficiency (EE%) through UPLC Analysis

The paclitaxel-coated and chitosan-modified selenium nanoparticles were evaluated using UPLC (ultra-performance liquid chromatography) method and the samples were passed through the C-18 column, with 10 µL injection volume with PDA detector, at RT, equipped with Empower 3 software. The flow rate of the sample was 1.0/min and the PTX drug detection peak was at 240 nm. The mobile phase selected was acetonitrile–methanol in the ratio of 60:40. The encapsulation efficiency of the drug PTX in PTX-chit-SeNPs was calculated according to Equation (1), as described in [25].
Encapsulation efficiency % =
(1)$$\frac{\;\mathrm{Total}\;\mathrm{PTX}\;\mathrm{used-amount}\;\mathrm{of}\;\mathrm{free}\;\mathrm{PTX}\;\mathrm{present}\;\mathrm{in}\;\mathrm{the}\;\mathrm{supernatant}}{\mathrm{Total}\;\mathrm{PTX}\;\mathrm{used}\;}\;\times \;100$$

### 2.6. Characterization of the Chitosan-Modified Selenium Nanoparticles Loaded with Paclitaxel

The confirmation of the paclitaxel-loaded chitosan-modified selenium nanoparticles was obtained using Fourier transform infrared spectroscopy (FTIR) by Shimadzu Spectroscopy (Model IR Affinity-1, which helped in the identification of the functional groups involved in the interactions of the paclitaxel with the modified selenium nanoparticles and was compared with paclitaxel, chitosan, and biosynthesized SeNPs that have a wavelength range from 400 to 4000 cm^−1^. The morphological analysis was conducted using transmission electron microscopy (TEM), where the samples were sputtered on the copper grids.

### 2.7. In Vitro Drug Release Analysis

The in vitro release of PTX from the nanosystem was studied using the dialysis method. In a dialysis bag (MWCO 60 kDa), 10 mL of nanosuspension containing PTX-chit-SeNPs was immersed in 100 mL of pH 7.4 and 5.5 phosphate buffer solution (PBS) and was dialyzed at 37 °C under mild agitation, thereby mimicking the physiological conditions. At a regular interval of time, 2 mL of the external solution was extracted and again replenished with 2 mL of fresh medium. The solution was centrifuged at 10,000 rpm for 15 min and the supernatant containing the free drug was evaluated for the concentration of PTX using a spectrophotometer. A standard graph was generated for PTX, where the R^2^ value was 98.85, and the drug concentration was calculated using slope and intercept values [26].

### 2.8. Cytotoxicity Studies of Paclitaxel-Loaded Chitosan-Modified Selenium Nanoparticles

The nanosystems-induced cytotoxicity was evaluated with the most commonly used MTT assay, where the viability percentage of the cells is measured when they transformed MTT to purple formazan precipitate. In a 96-well cell culture plate, 5000 cells/well were seeded, then incubated until 80% confluence was reached for 24 h at 37 °C. The next day, the spent media was removed and replaced with fresh media containing the treatment drugs at various concentrations (5 µg/mL to 100 µg/mL) for the biosynthesized SeNPs, SeNPs coated with chitosan, and PTX-chit-SeNPs. After incubation of 24 h, the media was again removed and MTT reagent (5 mg/mL) was added to each well and was kept for an incubation period of 4 h. The medium was then removed and replaced with 100 µL DMSO in each well, which was added to solubilize the crystalline formazan precipitate. The absorbance was recorded at 590 nm using an ELISA microplate reader. All the experiments were performed in triplicates and were expressed as the mean ± standard error [27].

### 2.9. Stimulation of Apoptosis by PTX-chit-SeNPs Compared with Biosynthesized SeNPs

The cell apoptosis was detected with FITC-Annexin V/PI (propidium iodide) double staining assay, according to the protocol (Invitrogen). The cells (1 × 10^5^) were seeded in 6- well culture plates and were incubated for 24 h at 37 °C. The cells were then treated with selected IC_50_ dose values for PTX-chit-SeNPs and biosynthesized SeNPs. The apoptotic and live cells were analysed with flow cytometry and were analysed with CytExpert 2.3 software.

### 2.10. Stimulation of Cell Cycle Arrest by PTX-chit-SeNPs Compared with Biosynthesized SeNPs

The cells (1 × 10^5^) were seeded in 6-well culture plates and were incubated for 24 h at 37 °C. The cells were supplemented with IC_50_ dose values for PTX-chit-SeNPs and biosynthesized SeNPs for 24 h. They were trypsinized and washed with 1x PBS, the obtained pellet was ethanol-fixed with chilled 70% ethanol solution for 2 h at low temperature. The pellet was washed to remove any traces of ethanol and was then incubated with the PI solution (1 mL) containing RNAase and PBS solution for 15 min at RT prior to analysis with a flow cytometer.

### 2.11. RNA Isolation and RT-PCR Assay of the Apoptotic Genes

The cells with a concentration of 1 × 10^5^ were seeded for 24 h at 37 °C in 6-well plates, then the cells that were treated with IC_50_ dose values for PTX-chit-SeNPs and biosynthesized SeNPs for 24 h were selected for the analysis. The cells were trypsinized, washed with 1x PBS, and then homogenized with triazole reagent (0.5 mL). Then, 0.2 mL of chloroform was added, and after 15 min incubation, the solution was centrifuged at 10,000 rpm for 10 min at 4 °C. The upper phase of transparent solution was transferred to fresh tubes, which were incubated with 0.5 mL isopropanol for 10 min at RT, followed by centrifugation. The RNA pellets obtained were dispersed in RNase-free water (0.1 mL) and washed with 75% ethanol. The dried RNA pellet was quantitively evaluated with a NanoDrop spectrophotometer. Then, 1 µg RNA samples were reverse-transcribed to synthesize cDNA samples and the cycling program was followed according to the manufacturer’s protocol given in the Primescript RT reagent kit (Takara). The apoptotic genes were selected, as shown in Table 1, and standard GAPDH-normalized with other genes.

The relative gene expression was calculated according to the given equation:Relative expression values = 2 (Ct GAPDH − Ct target gene)(2)
where Ct = Threshold level cycle [28].

### 2.12. Intracellular ROS Generation by PTX-chit-SeNPs Compared with Biosynthesized SeNPs

For the flow cytometer analysis, HeLa cells (1 × 10^5^) were cultured in 6-well plates for 24 h at 37 °C, after the incubation period, they were exposed to IC_50_ dose values for PTX-chit-SeNPs and biosynthesized SeNPs under similar conditions. The cells were trypsinized, washed with 1x PBS, and the resulting pellet was resuspended in PBS containing 10 µM DCFDA solution, followed by an incubation period of 30 min. The treated cells were analyzed using a flow cytometer and evaluated using CytExpert software.

### 2.13. Statistical Analysis

The experimental runs were made in triplicate, and the values were analyzed using the ANOVA statistical method. The results were statistically significant when the *p*-value was <0.05 and was expressed as the mean ± standard error.

## 3. Result and Discussion

### 3.1. Encapsulation Efficiency

The encapsulation efficiency was performed using UPLC analysis; it was observed that at 0.5 min, the absorbance peak for the standard PTX was at 0.393, which was almost similar to PTX-SeNPs-chit at 0.403, as shown in Figure 1. Therefore, the result signifies that there was a visible coating of the drug on the nanosystem. It can be assumed that the mechanism behind the encapsulation is that a hydrophobic drug, such as paclitaxel, interacts with a polymer, such as a chitosan, via hydrophobic–hydrophilic interactions, thereby increasing its solubility. The PTX molecule is composed of a hydrophobic region and a moderately hydrophilic region containing secondary amine and hydroxyl groups that can produce hydrogen bonds with chitosan molecules [20,29]. The encapsulation efficiency was calculated to be approximately 80%, according to the given equation.

### 3.2. Characterization of PTX-chit-SeNPs

The morphology and surface topography of the nanoparticles show a significant role in transportation across the cell membranes. These particles deliver the active therapeutic agent in a controlled and target-specific manner. According to the HR-TEM analysis in Figure 2A, the biosynthesized SeNPs are dispersed and spherical with an average size range of 98 nm. In the case of chitosan-modified SeNPs in Figure 2B; the chitosan moiety surrounded the spherical-shaped nanoparticles and the average size range was calculated to be 142.8 nm. While the size range for PTX-chit-SeNPs was 167 nm with spherical shape SeNPs coated with drug molecule and modified with chitosan (indicated through arrows) as shown in Figure 2C. The dark spherical form is the SeNPs, while the smaller round particles are PTX, which is covered with chitosan on the exterior. Similar results were observed for Siqi Zeng et al. [30] and Kaikai Bai et al. [31]. The spherical morphology of NPs can enable cellular internalisation. The smaller size range is possible due to either ionic or hydrophobic interactions between the polymer and the drug.

The major functional groups involved in the stabilization or reduction of SeNPs were −OH or carboxylic acids stretching, at 3294.78 cm^−1^; C=C with medium intensity, at 1628.73 cm^−1^; secondary alcohol, at 118.93 cm^−1^; and halogen compounds, at 717.8 cm^−1^, as shown in Figure 3D. The groups assigned for chitosan include amine, at 3356.92 cm^−1^; alkane, at 2886.43 cm^−1^; C=O stretching, at 1660.85 cm^−1^; aromatic, at 1591.10 cm^−1^; and strong C-O, at 1035.58 cm^−1^, as shown in Figure 3C. For PTX, the functional groups involved at 3474.15 cm^−1^ were −OH and −NH; at 3001.96 cm^−1^, alkenes or aromatic groups; at 2908.61 cm^−1^, alkanes −CH or –CO groups; at 1440.56 cm^−1^, aromatic groups; at 1309.91 cm^−1^, −CN; at 1034.62 cm^−1^, −CO; at 948.75 cm^−1^, −C-C; at 696.17 cm^−1^, alkene or aromatic; and at 511.86 cm^−1^, halogen compounds, as shown in Figure 3B. For PTX-chit-SeNPs, the functional groups assigned include 3319.27 cm^−1^ for −OH (alcohol or carboxylic acid) with strong and broad intensity, 2928.62 cm^−1^ and 2869.78 cm^−1^ with medium intensity for the alkane group, 1557.71 cm^−1^ for diketones, 1404.06 cm^−1^ for CH_2_ bending and CH_3_ deformation, 1016.32 cm^−1^ for −C-O stretching for alcohol or phenol groups, and 739.6 cm^−1^ for halogen compounds, as shown in Figure 3A. The FTIR analysis reports suggest that peaks distinctly available in PTX-chit-SeNPs are mainly due to the coating of individual components.

#### SeNPs

The EDAX analysis showed elements such as Se, Na, Cl, C, and O. The presence of Cl helped maintain the 3D integrity of the spherical structure, which was present at the end of Se chains and allowed the interaction between the chains. The presence of C and O was due to the polymer structure of the chitosan with maximum atomic weight. Se at peak 1.3 keV, Na at 1.04 keV, Cl at 2.621 keV, C at 0.277 keV, and O at 0.525 keV were in accordance with the energy table for EDS analysis (JEOL certificated), as shown in Figure 4.

Zeta potential offers information on the colloidal stability (> +30 mV or < −30 mV) and the electrostatic potential of particles in the reaction solution. The surface charge potential of the biosynthesized SeNPs was −23.4 mV. Still, the intensity changed with chitosan modification with +53.8 mV and the intensity further changed to +65.2 mV for PTX-loaded chitosan-modified SeNPs, as shown in Figure 5A–C. The interactions were mainly due to the ionic interactions between the positively charged PTX and negatively charged SeNPs. The hydrophobic–hydrophilic interaction between the chitosan and drug are mainly responsible for effective coating [32]. The positive charge is observed due to the protonated R-NH_3_+ form of the chitosan, which helps in the electrostatic interface of the R-NH3+ and the negative charge present on the cellular membrane or mucosal surfaces. Thereby, this interaction helps in the reversible structural rearrangement of the proteins of tight junctions, which benefits in the opening of these junctions and consequently assists in endocytosis [33].

The DLS analysis of the biosynthesized SeNPs confirmed that they are polydispersed with a hydrodynamic diameter of 101 nm. The hydrodynamic diameter of chitosan-modified SeNPs and PTX-loaded chitosan-modified SeNPs were found to be 156 nm and 172 nm, respectively which is almost according to TEM data (Figure 6). The average PI (poly-dispersion index) value of 0.33 was less than the standard 0.5, as shown in Figure 6A–C. The increase in particle size can be assumed to be due to the presence of electrostatic repulsion of the protonated amino groups and interchain hydrogen bonding attractions [34]. They remained in equilibrium at certain conditions, but with increasing chitosan concentration, there was a partial growth in the intermolecular cross-linking, thereby producing larger particles. The intermolecular cross-linking was reduced at decreased chitosan concentration, producing smaller particles [18].

### 3.3. In Vitro Drug Release Analysis

The release pattern of PTX was examined under pH conditions of 5.5 and 7.4 at 37 °C for 72 h in PBS solution. According to Figure 7, the release graph was dependent upon the pH of the medium. In the 5 h there was an increase in the release at pH 5.5 which continued to reach a maximum level of 92% for 72 h, while the maximum level remained 20% at pH 7. These results prove that the release of PTX can be at a maximum level when it is exposed to an acidic environment when compared to neutral pH. Through the process of endocytosis, the nanoparticles can easily be exposed to the acidic microenvironment and initiate a rapid release of the drug from the nanosystem, which can ultimately enhance the cytotoxic effect on cancer cells [35,36]. Therefore, the toxicity against normal cells can be minimized with negligible releases at physiological pH of 7.4. Similar results were observed when PTX was encapsulated in chitin nanoparticles, in which slow release of the drug was observed at neutral pH [37]. The effective release of PTX from the chitosan matrix at acidic conditions can also be explained as being due to the protonation of the amine groups, which causes a hydrophilization of the hydrophobic core or swelling of chitosan [38], and this eventually leads to the release of the drug [39].

### 3.4. In Vitro Cytotoxicity Analysis against HeLa Cell Line

The viability analysis was conducted for biosynthesized SeNPs, chitosan modified SeNPs, and PTX-chit-SeNPs against HeLa cell lines at concentrations within 5–100 µg/mL, as shown in Figure 8 and Figure 9A–D. The arrows indicate the fragmented apoptotic cells when exposed to the treatment nanosystems compared with control cells. There was a dose-dependent decrease in the inhibition rate at 24 h of incubation of treatment. The IC_50_ value considered for biosynthesized SeNPs, chitosan-modified SeNPs, and PTX-chit-SeNPs are 60 µg/mL, 45 µg/mL, and 30 µg/mL, and the viability rate decreased to approximately 20% for PTX-chit-SeNPs. The inhibition rate can be attributed to PTX assisting in the polymerization of microtubules [22], while the polymeric coating by chitosan enhances the cellular uptake of the hydrophobic drug via endocytosis [33], and the presence of SeNPs helps in the pro-oxidant activity, which helps in the generation of ROS, causing cell death [40]. The chitosan-modified SeNPs showed a similar effect on cytotoxicity, but it improved with the conjugation of PTX. Similar results were reported by T. Mary et al. [10,41] and J. Zang et al. [25].

### 3.5. Apoptosis Analysis for PTX-chit-SeNPs against HeLa Cell Lines

PTX stimulates the polymerization of the microtubule by binding to the β-subunit of the α/β-tubulin dimerin, which eventually restrained the growth of progressively dividing cells that result in programmed cell death or apoptosis [22]. Annexin V is a Ca^2+^- dependent protein with an affinity for the phospholipid phosphatidyl-serine, which forms cellular membranes. Dead cells undergo a translocation of this lining outward, which helps bind Annexin V protein labelled with FITC. PI stain distinguishes between viable and non-viable cells, and the stain can easily permeate through the plasma membrane of dead cells. The cell populations were differentiated as live, early, late, and necrotic or dead to quantify the rate of apoptosis [30,42]. HeLa cells were quantified for apoptosis analysis using PI and Annexin V-FITC dual staining to confirm this phenomenon. The apoptotic rate depended on the treatment dosage, as shown in Figure 10A–C. The early-stage and late-stage control cells with dual staining were observed to have cell populations of 8.77% and 0.05%, respectively. For the biosynthesized SeNPs at concentration 60 µg/mL, the population of cells was observed to be 29.35% at the early stage, and 10.90% at the later stage. The early and late stages for HeLa cells, when exposed to PTX-chit-SeNPs, were 47.61% and 6.50%, respectively. Therefore, the anticancer drug paclitaxel modified with chitosan on the biosynthesized SeNPs exhibited better ability to stimulate apoptosis in HeLa cells compared to the biosynthesized SeNPs without any surface modifications. The apoptotic characteristics such as nuclear condensation, chromatin condensation, or development of apoptotic bodies [41].

### 3.6. Cell Cycle Analysis of PTX-Chit-SeNPs Compared with Biosynthesized SeNPs

The cell cycle distribution was evaluated using CytExpert 2.3 software. DNA histograms were selected for analysing the population of cells in each phase, such SubG1, G0/G1, S, and G2/M. The apoptotic cell population containing only DNA content was measured by quantifying the sub-G1 peak, and 100,000 events were selected for each experiment, per sample. As observed in Figure 11, the data suggests that when the permeabilized cells were exposed to PTX-chit-SeNPs for 24 h, they showed a significant increase in the cell population in sub-G1 phase from 6.89% for control, 29.40% for biosynthesized SeNPs at 60 µg/mL concentration, and 49.35% for PTX-chit-SeNPs at 30 µg/mL concentration. The results suggest that PTX-chit-SeNPs induced cell death due to activation of apoptosis. Similar results were observed in [19].

### 3.7. Intracellular Investigation of ROS Analysis

The pro-oxidant effect and toxicity of the nanosystem against HeLa cells were examined when the cells were exposed to the nanosystem. The principle behind the nanosystem’s mechanism is that, when the cells are exposed to it antioxidant enzymes, such as phospholipid hydroperoxide, glutathione peroxidase, thioredoxin, etc. are induced to produce superoxide radicals which cause oxidative stress to the cells and may lead to cell death as a consequence of events such as DNA damage, cell cycle arrest, apoptosis, activation of caspase proteins, and so on [43,44]. According to Figure 12A, the control cells were treated with DCDFA and demonstrated a higher number of live cells (up to 99.81%), the biosynthesized SeNPs in Figure 12B were evaluated and demonstrated 41.63% live cells and 55.80% dead cells at concentration 60 µg/mL. The PTX-chit-SeNPs in Figure 12C, when exposed, demonstrated 27.25% live cells and 71.73% dead cells. The surface-decorated SeNPs with anticancer drug and polymer showed better results than green-synthesized SeNPs mediated through aqueous seed extract. Therefore, this combination can be utilized in the treatment of various diseases caused by oxidative stress, such as Leigh syndrome, cancer, diabetes, skin disease, and others [45].

### 3.8. Comparative Gene Expression Analysis of Biosynthesized SeNPs and PTX-chit SeNPs against Apoptotic Genes

The RT-PCR technique was used for analysing the gene expression of BAX and BCL-2 genes at a dosage of 60 µg/mL for biosynthesized SeNPs, and then at 30 µg/mL for PTX-chit-SeNPs, with a treatment period of 24 h. According to Figure 13, the PTX-chit-SeNPs significantly down-regulated the expression of BCL-2 and up-regulated the expression of BAX when compared to biosynthesized SeNPs. The treatment of PTX-chit-SeNPs is a predisposed mechanism in the apoptotic induction of chemoresistant cervical cells. The Bcl-2 protein family consists of both pro-apoptotic (Bax and Bak) and anti-apoptotic proteins (Bcl-2, Bcl-XL). The anti-apoptotic proteins inhibit the release of cytochrome c from mitochondria into the cytoplasm. In contrast, the pro-apoptotic proteins help release cytochrome c, activating the apoptotic cascade, and thereby inducing cell death. The pro-apoptotic protein BAX is one of the members of the BCL-2 family of apoptotic signalling proteins and an increase in the level of BAX has led to the enhancement of the rate of apoptosis [10,28]. In our investigation, the treatment with PTX-chit-SeNPs at a concentration of 30 µg/mL instigated apoptosis by increasing the expression in the Bax gene in HeLa cells after the incubation period of 24 h. Similar results were reported in [46].

## 4. Conclusions

The encapsulation of anticancer drugs such as paclitaxel into green-synthesized selenium nanoparticles modified with a hydrophilic biocompatible polymer, such as chitosan, and its synergistic effects against cervical cancer was demonstrated. The nanoformulation was characterized using various microscopic and spectroscopic approaches, with an average size of 170 nm. The successful loading of a hydrophobic anticancer drug, such as paclitaxel, along with targeted drug release analysis at pH 5.5, mimicking an acidic microenvironment, was also demonstrated. The mechanistic in vitro study analysis of PTX-chit-SeNPs was performed at a concentration of 30 µg/mL against HeLa cell lines. Based on these results, the synthesized nanosystem can be considered as a biocompatible, environment-friendly, and cost-effective anticancer therapeutic agent developed for cancer diagnostics. It can be used to improve human health.

## Figures and Tables

**Figure 1 molecules-27-07290-f001:**
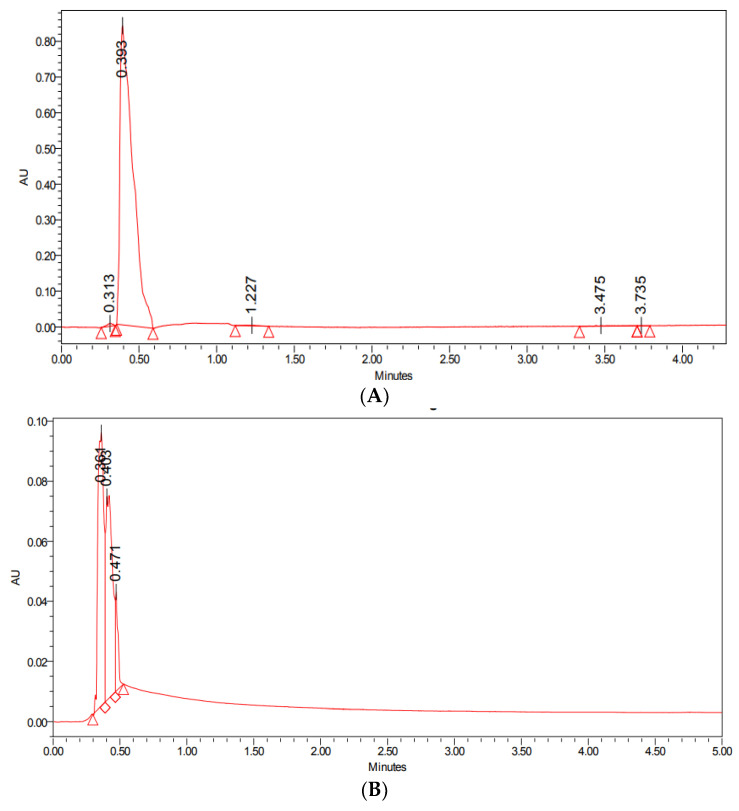
UPLC analysis of (**A**) standard drug paclitaxel, and (**B**) PTX-SeNPs-chit.

**Figure 2 molecules-27-07290-f002:**
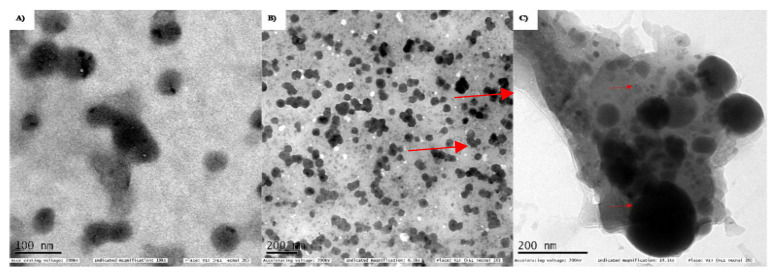
TEM analysis for (**A**) biosynthesized SeNPs using seed extract, (**B**) SeNPs modified with chitosan, (**C**) PTX (small rounds)-chit (covered as background)-SeNPs (larger spherical).

**Figure 3 molecules-27-07290-f003:**
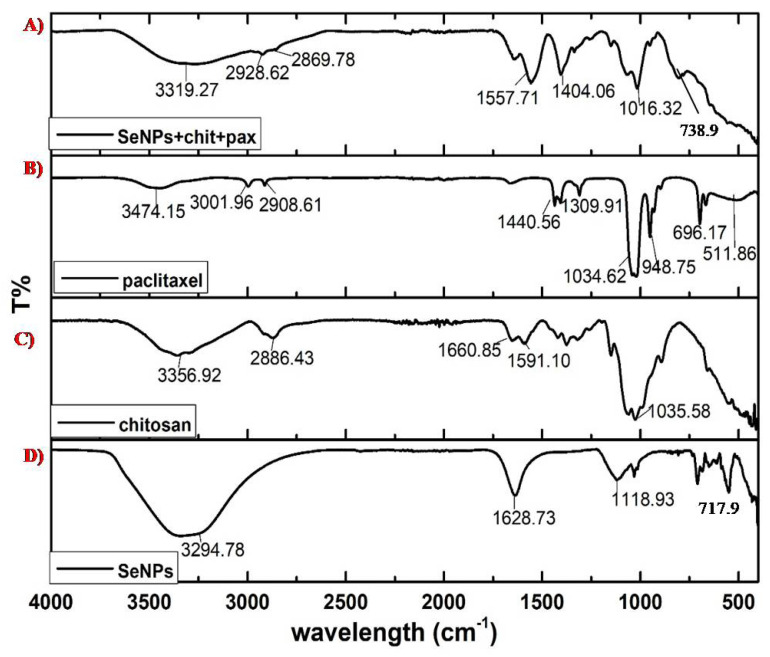
FTIR analysis of (**A**) SeNPs+chit+PTX, (**B**) PTX, (**C**) chitosan, and (**D**) biosynthesized.

**Figure 4 molecules-27-07290-f004:**
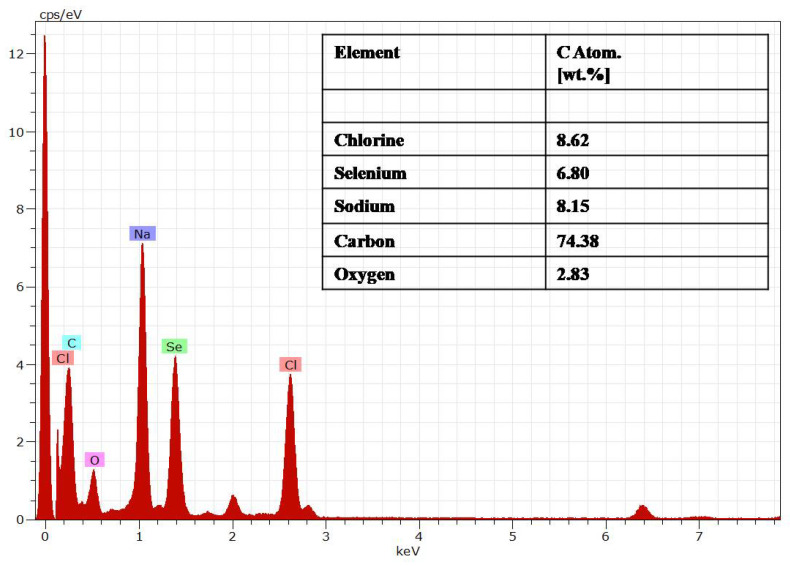
Energy diffraction spectroscopy (EDAX) of the PTX-chit-SeNPs.

**Figure 5 molecules-27-07290-f005:**
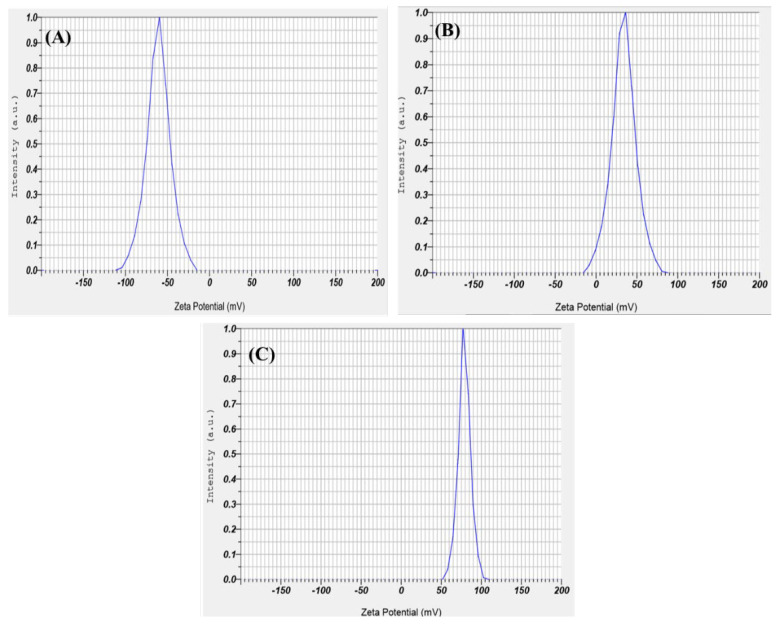
Zeta potential for (**A**) biosynthesized SeNPs using seed extract, (**B**) SeNPs modified with chitosan, and (**C**) PTX-loaded chitosan-modified SeNPs.

**Figure 6 molecules-27-07290-f006:**
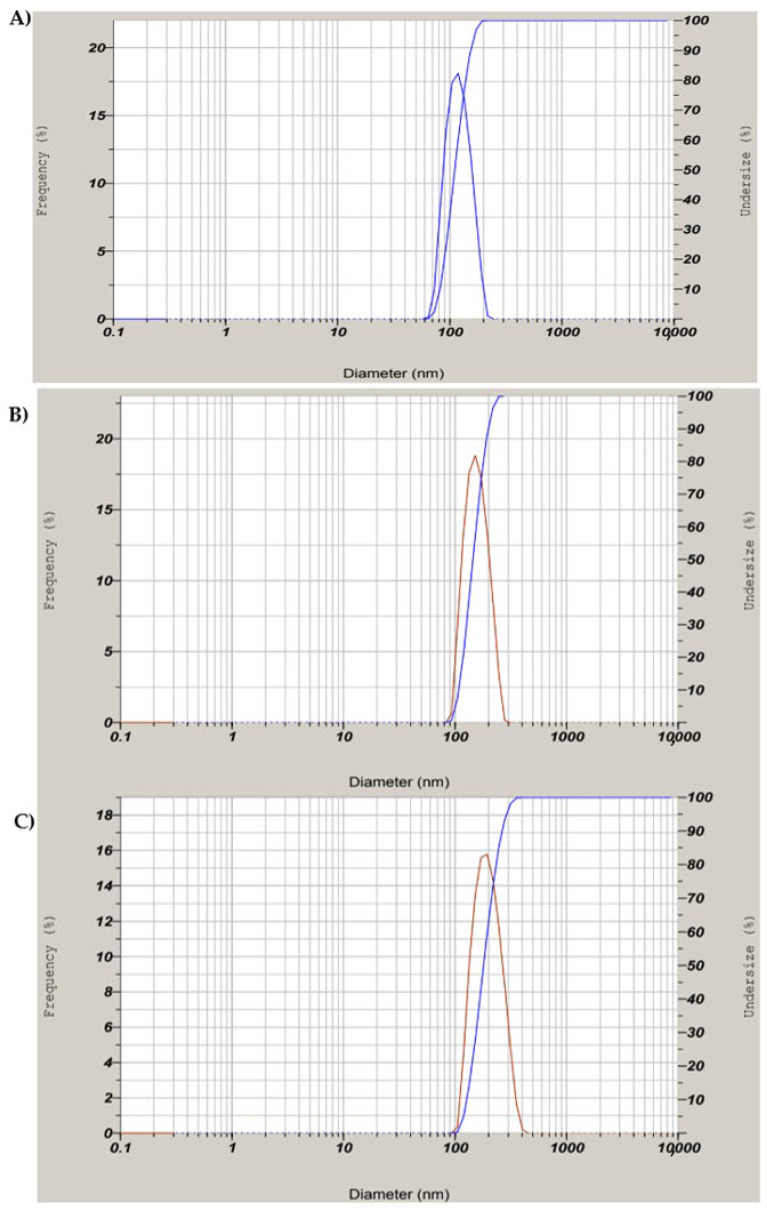
DLS analysis for (**A**) biosynthesized SeNPs using seed extract, (**B**) SeNPs modified with chitosan (**C**) PTX-loaded chitosan-modified SeNPs.

**Figure 7 molecules-27-07290-f007:**
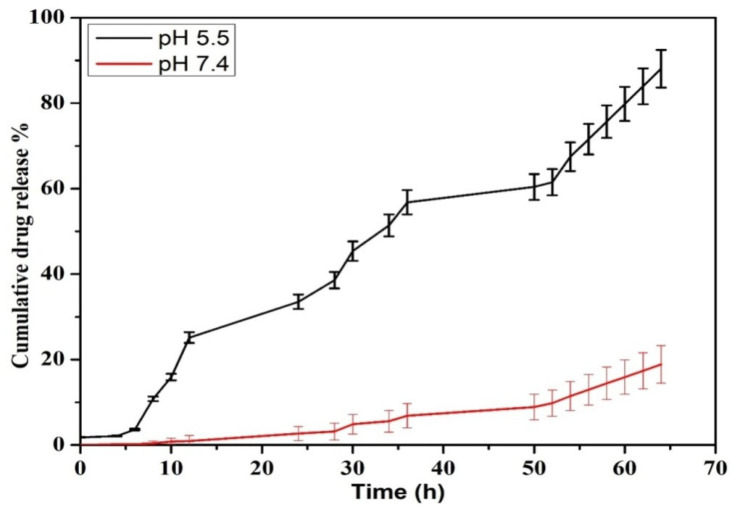
Drug release kinetics for PTX released from PTX-chit-SeNPs.

**Figure 8 molecules-27-07290-f008:**
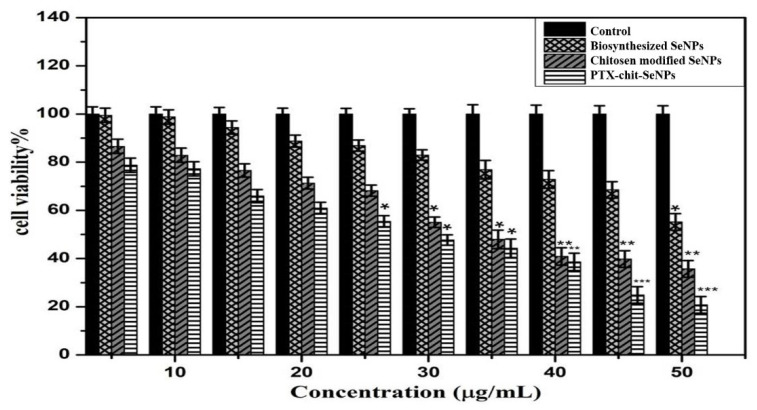
MTT analysis for biosynthesized SeNPs, chitosan-modified SeNPs, and PTX-chit-SeNPs against HeLa cell lines. Data are represented as mean ± SD for experiment in triplicate * indicates *p* < 0.05 versus control, ** indicates *p* < 0.01, and *** indicates *p* < 0.001.

**Figure 9 molecules-27-07290-f009:**
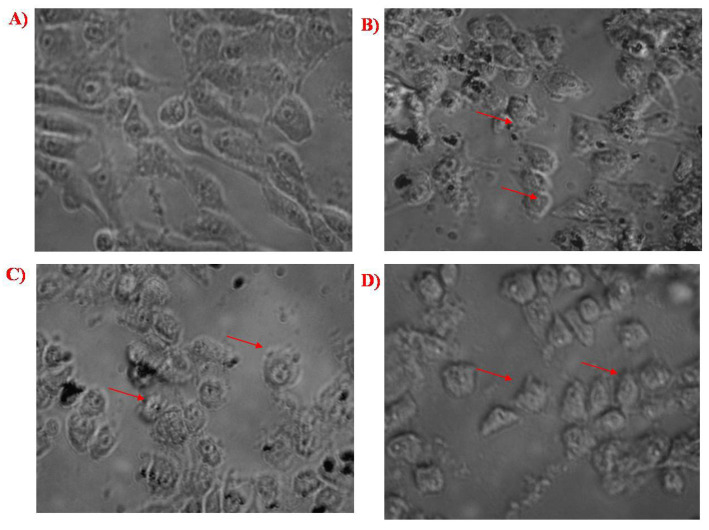
(**A**) Control cells, (**B**) cells treated with biosynthesized SeNPs, (**C**) cells treated with chitosan-modified SeNPs, and (**D**) cells treated with PTX-chit-SeNPs.

**Figure 10 molecules-27-07290-f010:**
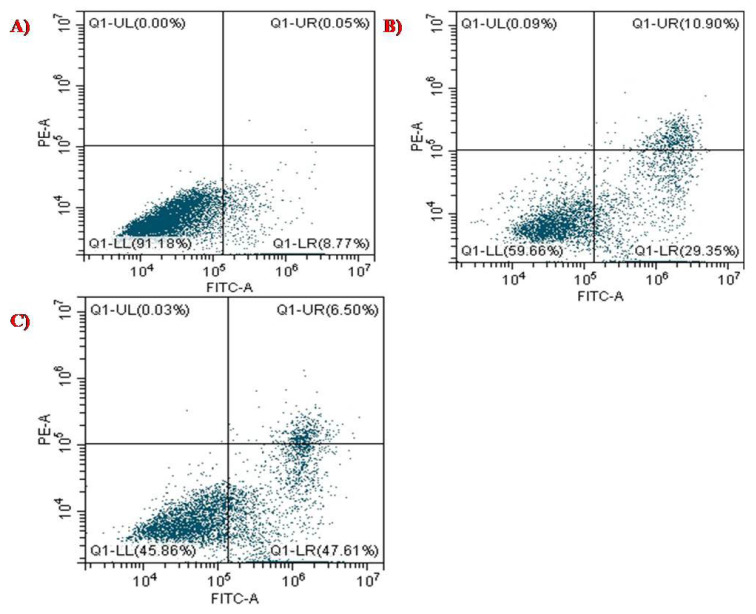
Apoptosis analysis of (**A**) control with dual staining of FITC+PI, (**B**) biosynthesized SeNPs at 60 µg/mL concentration, and (**C**) PTX-chit-SeNPs at 30 µg/mL concentration against HeLa cell lines.

**Figure 11 molecules-27-07290-f011:**
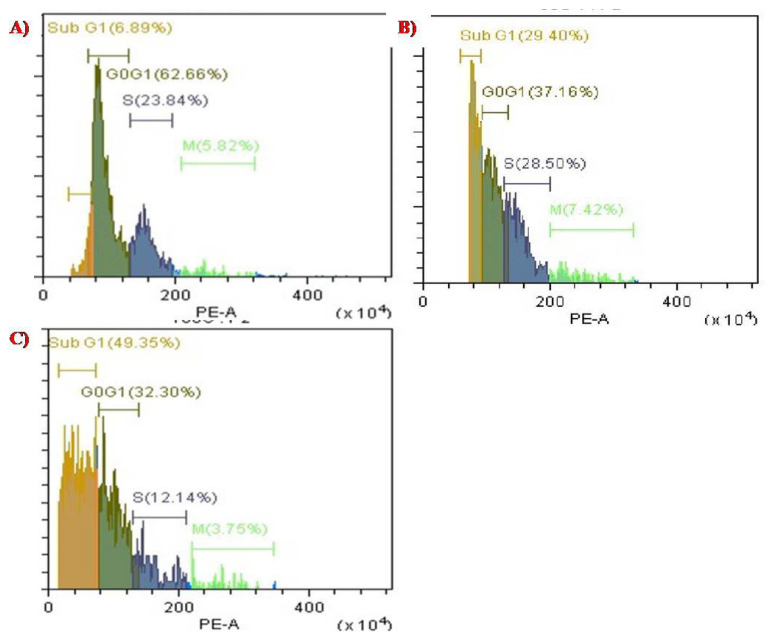
Cell cycle analysis of (**A**) control with PI stain, (**B**) biosynthesized SeNPs at 60 µg/mL concentration, and (**C**) PTX-chit-SeNPs at 30 µg/mL concentration.

**Figure 12 molecules-27-07290-f012:**
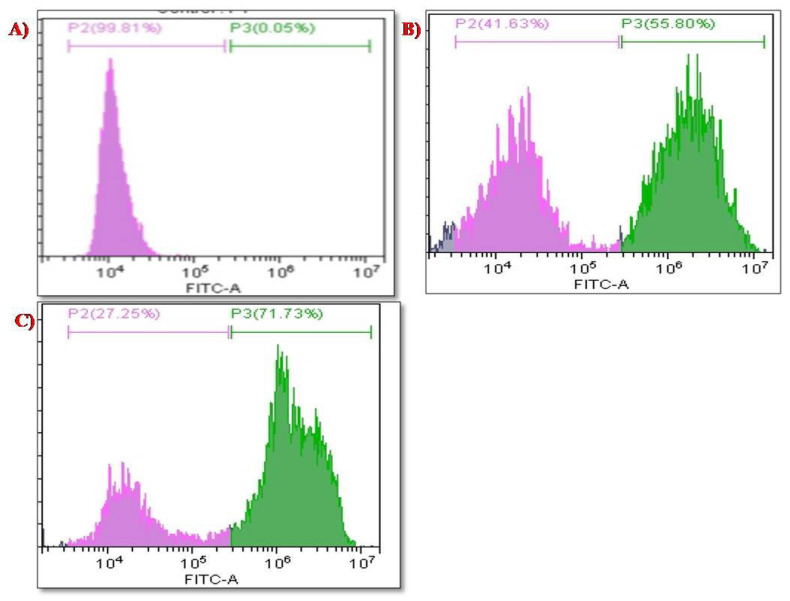
ROS analysis for (**A**) control cell population, (**B**) biosynthesized SeNPs, (**C**) PTX-chit-SeNPs-treated population.

**Figure 13 molecules-27-07290-f013:**
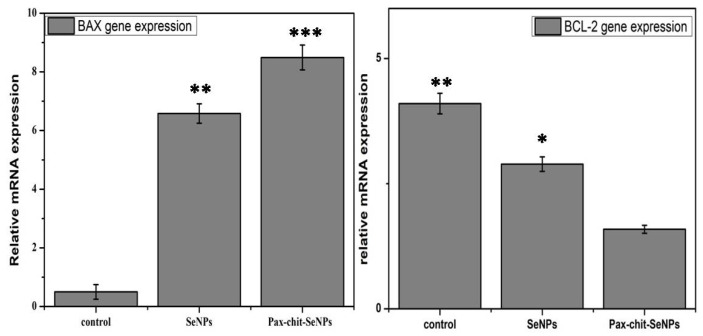
BAX and Bcl-2 gene expression normalized with GAPDH when treated with biosynthesized SeNPs at 60 µg/mL, and PTX-chit-SeNPs at 30 µg/mL. Data are represented as mean ± SD for experiment in triplicate; * indicates *p* < 0.05 versus control, ** indicates *p* < 0.01, *** indicates *p* < 0.001.

**Table 1 molecules-27-07290-t001:** Forward and Reverse primers of apoptotic genes used for RT-PCR.

Genes	Forward Primer (5′-3′)	Reverse Primer (5′-3′)
Bcl_2_	5′-CTTTTGCTGTGGGGTTTTGT-3′	5′-GTCATTCTGGCCTCTCTTGC-3′
Bax	5′-GGAGCTGCAGAGGATGATTG-3′	5′-CCTCCCAGAAAAATGCCATA-3′
GAPDH	5′-GAAGGTGAAGGTCGGAGT-3′	5′-GAAGATGGTGATGGGATTTC-3′

## Data Availability

Not applicable.

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
