# Peer review of "Preparation of Paclitaxel-Encapsulated Bio-Functionalized Selenium Nanoparticles and Evaluation of Their Efficacy against Cervical Cancer"

_molecules, 2022, doi:10.3390/molecules27217290_

Round 1

Reviewer 1 Report

S. Menon et al. have presented a bio-functionalization of Se NPs (previously reported by the group) with chitosan and Paclitaxel binding, to produce nanoparticle systems able to release the drug at relevant pH values (7.4, 5.5) and thus can be utilized against cancer cells (MTT analysis, apoptosis, gene expression analysis, cell cycle arrest, and over-production of ROS).

Regarding the title (more specifically, the first word): "created and performed spontaneously or without preparation; impromptu". Is this the right word to use (“Improvised”) regarding the anticancer potential of a known drug? Same observation regarding the conclusion.

Line 40 – “without causing any side effects” – that is at least debatable.

Line 89 : While presenting each part of the final  NP system is useful, another phrase linking these notions together should be included, expressing why the authors chose this combination and the envisioned advantages.

Line 94: analytically graded => analytical grade

Line 103: by Takara Bio => from …

Lines 111-112: “Further drop-wise addition of paclitaxel (dissolved in DMSO) to the chitosan-coated selenium nanoparticles” – needs rephrasing.

Line 121: units for flow rate missing

The correct abbreviation of the therapeutic drug used (Paclitaxel) is PTX [source: https://www.cancer.gov/publications/dictionaries/cancer-drug/def/paclitaxel ]. This should also be consistent (“Pax”, “PAX” etc encountered throughout the draft)

Line 215: reference needs to be added for said equation. (add numbers to equations, specifically to the efficiency relation at line 126)

Figure 1: resolution is faily poor and overlapping occurs on Fig.1B, therefore this figure needs to be redrawn.

Line 229: observed for => should be “by”

Fig. 2. IS that HR-TEM? There are SEM microscopes with higher resolution. Please check scale and correct as necessary.

The IR data , while very detailed, may need a second look; for instance, the peak at 1680 could be C=O stretching etc.

The language needs to be checked throughout the text, for instance on line 285: “were in the range of 172 nm, which is almost according to the obtained optimized results.”

Have the authors run the release profile assessment of PTX from the PTX/chit/Se NPs at least twice? The profile seems a bit off, although the overall trend is expected. (Fig. 7, line 317)

The draft is well written and after a minor revision step can be published in Molecules.

Author Response

Manuscript ID: molecules-1952781

Dear Editor and reviewers

Thank you very much for your kind consideration and invaluable efforts to review our manuscript. We are truly grateful for the meaningful comments and insights of the reviewers and editor. We have carefully revised all issues raised by the reviewers. We feel that our manuscript has been significantly improved from the reviewer’s creative and insightful suggestions. We would like to submit our fully revised manuscript after revised of all comments. All corrections made are highlighted with green color in the revised manuscript. Please see the response to reviewers’ comments on the succeeding pages.

Thank you very much for your consideration.

Sincerely Yours,

Nagaraj Basavegowda

Reviewer 1

Comment 1

Regarding the title (more specifically, the first word): "created and performed spontaneously or without preparation; impromptu". Is this the right word to use (“Improvised”) regarding the anticancer potential of a known drug? Same observation regarding the conclusion

Response

We appreciate the reviewer for their valuable suggestion regarding the Title of the manuscript. As per the comment, we have modified the title as follows

Title: Preparation of Paclitaxel-Encapsulated Bio-Functionalized Selenium Nanoparticles and Evaluation of their Efficacy against Cervical Cancer

Comment 2A

Line 40 – “without causing any side effects” – that is at least debatable

Response

We appreciate the reviewer for their valuable suggestion. As per the reviewer’s comment, we have modified the manuscript as follows

Line 40: “with negligible toxicity”.

Comment 2B

Line 89: While presenting each part of the final  NP system is useful, another phrase linking these notions together should be included, expressing why the authors chose this combination and the envisioned advantages.

Response

We appreciate the reviewer for their valuable suggestion. As per the reviewer’s comment, we have modified the manuscript as follows. Kindly see below or Line No. 44-48, 70-77, 87-88, and 97-104 in the revised manuscript

Moreover, nanoparticles have high surface-area-to-volume proportions [1], yielding high adsorbing properties [2]. On this basis, nanoparticles are adsorbed with therapeutic drugs [3], phytochemicals with anticancer properties [4], imaging agents [5], or target-specific genes or peptides as targeting ligands to the cancer receptor cells [6].

But, the major problem associated with the application of SeNPs as anticancer activity is reported undesired toxicity against healthy cells [10]. Thus, we need to overcome this undesired toxicity without compromising its anticancer activity. The functionalization of the NP surface with biopolymers is one of the best-reported techniques to improve biocompatibility with reduced aggregation.

Thus, Chitosan is used for the functionalization which is a cationic hydrophilic linear polysaccharide composed of the acetylated unit (N-acetyl- d-glucosamine) and deacetylated unit of (β- (1–4)-linked D-glucosamine).

Moreover, SeNPs might also be used for drug delivery applications, thereby provid-ing a synergistic therapeutic effect against cancer.

Alternative formulations that can aid in improvising the effective delivery of PTX to tumor cells with low toxicity to the healthy cells like nanoparticles, liposomes, self-emulsions, and micelles. Among these, nanoparticles that are functionalized with phytochemicals from seed extract as novel agents can provide a promising delivery system with the ad-vantage of the EPR effect for passive targeting of PTX. The results of this study suggest that this nanocombinational system can be a promising alternative for cancer treatment with low toxicity, but, further analysis of the in-vivo or ex-vivo studies can still be reviewed in future studies.

Comment 3

Line 94: analytically graded => analytical grade

Response

We have corrected the manuscript in line No. 109

Comment 4

Line 103: by Takara Bio => from …

Response

We have corrected the manuscript in line 118.

Comment 5

Lines 111-112: “Further drop-wise addition of paclitaxel (dissolved in DMSO) to the chitosan-coated selenium nanoparticles” – needs rephrasing

Response

As per the advice of the reviewer, we have modified the manuscript as follows, kindly see below or line No. 133-141 in the revised manuscript

The chitosan solution of 1% concentration (w/v) was prepared in glacial acetic acid where the solution was kept in a magnetic stirrer for 2 h until it was completely dissolved. followed by the addition of 1% of the biosynthesized SeNPs with continuous stirring at RT for 10-12 h. After the12 h of stirring, paclitaxel in DMSO (at the required concentration) was added drop-wise at 1mL/min to the chitosan-coated selenium nanoparticles and kept for an overnight continuous stirring. The PTX-chit-SeNPs were then centrifuged at 10,000 rpm for 15 min, the pellet obtained was washed with Mili-Q water and the purified solution was then lyophilized. The moisture-free samples were then stored at Room Temperature for further analysis and characterization.

Comment 6

Line 121: units for flow rate missing

Response

We have mentioned the flow rate for the addition of paclitaxel in the manuscript (1 mL/min).

Comment 7

The correct abbreviation of the therapeutic drug used (Paclitaxel) is PTX [source: https://www.cancer.gov/publications/dictionaries/cancer-drug/def/paclitaxel]. This should also be consistent (“Pax”, “PAX” etc encountered throughout the draft)

Response

As per the advice of the reviewer, we have changed “Pax” to “PTX” throughout the manuscript.

Comment 8

Line 215: reference needs to be added for the said equation. (add numbers to equations, specifically to the efficiency relation at line 126)

Response

As per the advice of the reviewer, we have added equation numbers throughout the manuscript.

Comment 9

Figure 1: resolution is fairly poor and overlapping occurs on Fig.1B, therefore this figure needs to be redrawn

Response

As per the advice of the reviewer, we have replaced Figure 1 with a higher resolution. 

Kindly see below or Figure 1 in the revised manuscript

Comment 10

Line 229: observed for => should be “by”

Response

We have rectified this as per the reviewer’s advice in the revised manuscript

Comment 11

Fig. 2. Is that HR-TEM? There are SEM microscopes with higher resolution. Please check the scale and correct it as necessary.

Response

We thank the reviewer for pointing out the error. We have rectified in the revised manuscript as TEM.

Comment 12

The IR data, while very detailed, may need a second look; for instance, the peak at 1680 could be C=O stretching, etc.

Response

As per the advice of the reviewer, we have corrected the manuscript as follows, kindly see below or Line No. 267-268

2886.43 cm-1 for the alkane group, 1660.85 cm-1 for C=O stretching, 1591.10 cm-1 is assigned to an aromatic group, 1035.58 cm-1 is assigned to the strong C-O group Figure 3C. In the case

Comment 13

The language needs to be checked throughout the text, for instance on line 285: “were in the range of 172 nm, which is almost according to the obtained optimized results.”

Response

As per the advice of the reviewer, we have rephrased the entire phase, kindly see below or Line No. 309-312 in the revised manuscript

The DLS analysis of the biosynthesized SeNPs shows the polydispersed with a hydrody-namic diameter of 101 nm. The hydrodynamic diameter of chitosan-modified SeNPs found as 156 nm and 172 nm, respectively which is almost according to TEM data (Fig.6).

Comment 14

Have the authors run the release profile assessment of PTX from the PTX/chit/Se NPs at least twice? The profile seems a bit off, although the overall trend is expected. (Fig. 7, line 317)

Response

The release kinetic study was performed thrice and average values were used for plotting. We have mentioned in the Figure Legend.

Reviewer 2 Report

In this paper the synergistic effects of the chitosan-modified selenium nanoparticles loaded with paclitaxel (Pax-chit-SeNPs) were studied. In general, the presented research topic, the proposed methods and the description of the research results are presented in an interesting form. However, the authors should pay attention to improving the following points:

1.     Literature references [1-6] in the first paragraph should be more closely aligned with the description

2.     In the section Introduction lacks information about the novelty of the presented work in comparison with the solutions described so far in the literature

3.     In section 2.3, the Authors do not describe the biosynthesis of SeNPs but refer to a previous publication. A brief description of the synthesis should be added

4.     In section 2.5, please add the equation number

5.     Figure 1 - axis signatures and values marked in the figure are illegible

6.     Line 268 reference (Voigt et al., 2014) should be cited in the same way as in the rest of the article

7.     The signature to the Y axis in the Figure 6 is not visible

8.     Line 339 "*p" instead of "*P"

9.     After Figure 9 there is a wrong numbering of figures : line 365 “Figure” 10 instead of “Figure 1”, line 382 “Figure 11” instead of “Figure 2”, line 402 “Figure 12” instead of “Figure 3”

10.  The most important conclusions should be pointed out

Author Response

Manuscript ID: molecules-1952781

Dear Editor and reviewers

Thank you very much for your kind consideration and invaluable efforts to review our manuscript. We are truly grateful for the meaningful comments and insights of the reviewers and editor. We have carefully revised all issues raised by the reviewers. We feel that our manuscript has been significantly improved from the reviewer’s creative and insightful suggestions. We would like to submit our fully revised manuscript after revised of all comments. All corrections made are highlighted with green color in the revised manuscript. Please see the response to reviewers’ comments on the succeeding pages.

Thank you very much for your consideration.

Sincerely Yours,

Nagaraj Basavegowda

Reviewer 2

Comment 1

Literature references [1-6] in the first paragraph should be more closely aligned with the description

Response

We have rewritten the first paragraph as per the reviewer’s advice in the manuscript as follows, kindly see below or Line No. 44-48 in the revised manuscript  

Moreover, nanoparticles have high surface-area-to-volume proportions [1], yielding high adsorbing properties [2]. On this basis, nanoparticles are adsorbed with therapeutic drugs [3], phytochemicals with anticancer properties [4], imaging agents [5], or target-specific genes or peptides as targeting ligands to the cancer receptor cells [6].

Comment 2

In the section, Introduction lacks information about the novelty of the presented work in comparison with the solutions described so far in the literature

Response

We have significantly modified the introduction stating the importance of the current work as follows in the manuscript, kindly see below or Line No. 70-77, 87-88 and 97-104 in the revised manuscript

But, the major problem associated with the application of SeNPs as anticancer activity is reported undesired toxicity against healthy cells [10]. Thus, we need to overcome this undesired toxicity without compromising its anticancer activity. The functionalization of the NP surface with biopolymers is one of the best-reported techniques to improve biocompatibility with reduced aggregation.

Thus, Chitosan is used for the functionalization which is a cationic hydrophilic linear polysaccharide composed of the acetylated unit (N-acetyl- d-glucosamine) and deacetylated unit of (β- (1–4)-linked D-glucosamine).

Moreover, SeNPs might also be used for drug delivery applications, thereby providing a synergistic therapeutic effect against cancer.

Alternative formulations that can aid in improvising the effective delivery of PTX to tumor cells with low toxicity to the healthy cells like nanoparticles, liposomes, self-emulsions, and micelles. Among these, nanoparticles that are functionalized with phytochemicals from seed extract as novel agents can provide a promising delivery system with the ad-vantage of the EPR effect for passive targeting of PTX. The results of this study suggest that this nanocombinational system can be a promising alternative for cancer treatment with low toxicity, but, further analysis of the in-vivo or ex-vivo studies can still be reviewed in future studies.

Comment 3

In section 2.3, the Authors do not describe the biosynthesis of SeNPs but refer to a previous publication. A brief description of the synthesis should be added

Response

As per the advice of the reviewer, we have added the protocol for the biosynthesis of SeNPs in the manuscript, kindly see below or Line No. 121-131 in the revised manuscript.

In brief, the seeds were double washed with double distilled water (Mili-Q) shade dried for 5-6 days and the dried seeds were crushed, weighed and then added to 100ml of Mili-Q. The liquid was kept in the water bath at 80℃ for 45min, and filtered using Whatman filter paper 42. The filtrate was stored at 4℃ for further use. A ratio of 9:1 was maintained for precursor and extract concentration. The reaction continued till the colorless solution changed to the dark brown reaction mixture. The synthesized nanoparticles were further purified with double distilled water and ethanol solution, this helps in the removal of excess phytochemicals or untreated sodium selenite from the SeNPs solution. The pellet was then dialyzed against distilled water for a period of 12h. The purified nanoparticles were lyophilized for further characterization and storage.

Comment 4

In section 2.5, please add the equation number

Response

As per the advice of the reviewer, we have added equation numbers throughout the manuscript.

Comment 5

Figure 1 - axis signatures and values marked in the figure are illegible

Response

As per the advice of the reviewer, we have replaced Figure 1 with a higher resolution with better readability of values

Comment 6

As per the advice of the reviewer, the reference was removed as it was misplaced in the wrong text.

Response

As per the advice of the reviewer, the reference was removed as it was misplaced in the wrong text.

Comment 7

The signature to the Y axis in Figure 6 is not visible

Response

As per the advice of the reviewer, we have replaced Figure 6 with a higher resolution with better readability of Y axis.

Comment 8

Line 339 "*p" instead of "*P"

Response

We have checked for unwanted capitalization of words throughout the manuscript and corrected them.

Comment 9

After Figure 9 there is a wrong numbering of figures: line 365 “Figure” 10 instead of “Figure 1”, line 382 “Figure 11” instead of “Figure 2”, line 402 “Figure 12” instead of “Figure 3”

Response

We have corrected the figure numbers in the manuscript.

Comment 10

The most important conclusions should be pointed out.

Response

As per the advice of the reviewer, we have edited the text in the manuscript. Kindly see below or Line No. 456-467 in the revised manuscript

The encapsulation of anticancer drugs like paclitaxel into green synthesized selenium nanoparticles modified with a hydrophilic biocompatible polymer like chitosan and its synergistic effects against cervical cancer has been demonstrated. The nanoformulation was characterized using various microscopic and spectroscopic approaches, with an average size of 170 nm. The successful loading of a hydrophobic anticancer drug like paclitaxel along with targeted drug release analysis at pH 5.5 mimicking the acidic microenvironment was also demonstrated. The mechanistic in-vitro study analysis of PTX-chit-SeNPs was performed at a concentration of 30 µg/mL against HeLa cell lines.Based on these results, the synthesized nanosystem can be considered as a biocompatible, environment-friendly and cost-effective anticancerous therapeutic agent developed for cancer diagnostics. It can be used to improvise human health.
